# Nurses' continuance intention to use electronic health record systems: The antecedent role of personality and organisation support

Adi Alsyouf[1,2,3]*, Nizar Alsubahi[4,5]*, Haitham Alali[6], Abdalwali Lutfi[7,8], Khalid Anwer Al-Mugheed[9], Mahmaod Alrawad[10,11], Mohammed Amin Almaiah[12], Rami J. Anshasi[13], Fahad N. Alhazmi[4], Disha Sawhney[14]

1 Faculty of Business Rabigh, Department of Managing Health Services & Hospitals, College of Business (COB), King Abdulaziz University, Jeddah, Saudi Arabia, 2 Applied Science Research Center, Applied Science Private University, Amman, Jordan, 3 MEU Research Unit, Middle East University, Amman, Jordan, 4 Faculty of Economics and Administration, Department of Health Services and Hospitals Administration, King Abdulaziz University, Jeddah, Saudi Arabia, 5 Faculty of Health, Department of Health Services Research, Care and Public Health Research Institute—CAPHRI, Maastricht University Medical Center, Medicine and Life Sciences, Maastricht University, Maastricht, The Netherlands, 6 Faculty of Medical and Health Sciences, Health Management Department, Liwa College, Abu Dhabi, UAE, 7 College of Business Administration, The University of Kalba, Kalba, Sharjah, United Arab Emirates, 8 Jadara University Research Center, Jadara University, Irbid, Jordan, 9 College of Nursing, Riyadh Elm University, Riyadh, Saudi Arabia, 10 Quantitative Method, College of Business Administration, King Faisal University, Al-Ahsa, Saudi Arabia, 11 College of Business Administration and Economics, Al-Hussein Bin Talal University, Ma'an, Jordan, 12 Department of Computer Science, King Abdullah the II IT School, The University of Jordan, Amman, Jordan, 13 Faculty of Dentistry, Prosthodontics Department, Jordan University of Science and Technology, Irbid, Jordan, 14 Department of COO, Temple University Health System (Fox Chase Cancer Center), Philadelphia, PA, United States of America

* oal@kau.edu.sa (AA); nizar.alsubahi@maastrichtuniversity.nl (NA)

## Abstract

Nurses play a crucial role in the adoption and continued use of Electronic Health Records (EHRs), especially in developing countries. Existing literature scarcely addresses how personality traits and organisational support influence nurses' decision to persist with EHR use in these regions. This study developed a model combining the Five-Factor Model (FFM) and the Unified Theory of Acceptance and Use of Technology (UTAUT) to explore the impact of personality traits and organisational support on nurses' continuance intention to use EHR systems. Data were collected via a self-reported survey from 472 nurses across 10 public hospitals in Jordan and analyzed using a structural equation modeling approach (Smart PLS-SEM 4). The analysis revealed that personality traits, specifically Openness, Experience, and Conscientiousness, significantly influence nurses' decisions to continue using EHR systems. Furthermore, organisational support, enhanced by Performance Expectancy and Facilitating Conditions, positively affected their ongoing commitment to EHR use. The findings underscore the importance of considering individual personality traits and providing robust organisational support in promoting sustained EHR usage among nurses. These insights are vital for healthcare organisations aiming to foster a conducive environment for EHR system adoption, thereby enhancing patient care outcomes.

**Data Availability Statement:** All relevant data are within the paper and its Supporting Information files.

**Funding:** This research was funded through the annual funding track by the Deanship of Scientific Research, from the vice presidency for graduate studies and scientific research, King Faisal University, Saudi Arabia [KFU241359].

**Competing interests:** The authors have declared that no competing interests exist.

## 1. Introduction

Healthcare systems face complex costs, increased demands, inadequate and low-quality care, and inefficient care processes [1]. Governments are adopting several solutions to address these problems, such as information technology (IT) in healthcare [1–4]. Health information technology (HIT) is an umbrella term referring to a wide variety of goods and services, including telehealth services [5–8], assistive technology (AT) and sensors [9–12], electronic health records (EHR) [13–16], personal health records [3,17,18], mobile healthcare technologies [19–26], tele-monitoring tools, and telehealth [5–8], wearable health devices [27]. These technologies allow all healthcare professionals to gather, share, and use health information.

One of the most sophisticated tools that can be deployed to store and process health information using health information technology is the Electronic Health Record (EHR), a tool that can significantly improve the efficiency and quality of service delivery in this regard [28–30]. HIT deployments are growing and becoming a crucial instrument for enhancing the effectiveness of services in this sector as a result [30].

According to previous studies, EHR offers many advantages when considering organisational, social, and clinical implications [31]. These include lower research costs, enhanced safety [31] and quality of care [32], accelerating medical discoveries [33], quickly filtering information as needed, preventing and reducing errors related to medical care, enabling easy reading of information, and can be completed devoid of the problems associated with handwriting, and offering various healthcare delivery options, exchange data between the parties [13,16,34].

Despite these advantages, EHR systems, like other systems, have shortcomings. The implementation and acceptance of such systems may involve issues of system productivity and compatibility, quality and security [35–37], balancing costs and benefits [38], interoperability, technology and reimbursement issues, confidentiality issues [39,40], readiness and regulatory issues, and integration and customisation issues [37,41,42].

The adoption and implementation of health information systems (HIS) often encounter behavioral issues such as non-use, resistance, or even rejection by users, leading to the "shelfware" problem where new technologies remain underutilized [43,44]. Initial acceptance of technologies like electronic patient files does not necessarily translate into their continued use, as users may reevaluate their decisions due to various factors [13,16]. These behavioral challenges have led to the failure or shelving of numerous health industry implementation initiatives, partly due to a lack of understanding of how healthcare professionals react to installed HIT [43,45,46].

This is particularly relevant in the context of nurses, which represent the majority of employees in healthcare organizations and are integral to the operational success of these systems. Nurses are at the forefront of utilizing EHR systems in their daily practice, directly influencing the efficiency and effectiveness of healthcare delivery. Their attitudes and behaviors toward EHR systems are crucial in determining the success of these technologies in clinical settings. Despite their importance, there is a noticeable gap in the literature regarding the post-adoption behavior of nurses, especially in developing countries like Jordan.

Building on this understanding, this study delves into the role of personality traits and organizational support in influencing the continuance intention of nurses to use EHRs. Personality traits, as per the Five-Factor Model (FFM), significantly influence healthcare professionals' behavior, shaping their attitude towards technology adoption and affecting their willingness to continue using EHR systems [13,47,48]. Organizational support, particularly from top management, plays a vital role in reducing uncertainties and fostering a conducive environment for the sustained adoption and effective use of EHR systems within healthcare settings [13,16,49–53].

This study delves into the post-adoption behavior of nurses in Jordan, examining how their individual personality traits and organizational support influence their sustained engagement with Electronic Health Records (EHRs). It seeks to untangle the intricate relationship between personal dispositions and organizational context in driving long-term use of EHR systems. This research is crucial in the realm of healthcare technology adoption, as it not only investigates the impacts of individual personality factors but also examines the role of organizational support mechanisms. It addresses the critical need for healthcare organizations to enhance patient care and operational efficiency by fostering effective EHR usage among nurses. Fostering effective usages is particularly important for healthcare management and policymakers, and the study provides insights into optimizing EHR implementation strategies that are attuned to nurses' personality traits and the organizational support available. Furthermore, this research fills a significant gap in empirical studies on EHR post-adoption phases, especially in the context of developing countries like Jordan. Previous research has overlooked the combined influence of personality traits and organizational support on nurses' continued use of EHRs, ignoring the vital interplay between internal individual factors and external organizational dynamics. By proposing an innovative model that integrates the Five-Factor Model (FFM) with the Unified Theory of Acceptance and Use of Technology (UTAUT), this study aims to offer a comprehensive understanding of how personality and organizational support jointly influence nurses' decision to persist with EHR use, thereby contributing to more effective technology implementation and utilization in healthcare settings.

## 2. Literature review

### 2.1 EHR adoption

The adoption of Electronic Health Records (EHRs) is a crucial aspect of health information technology (HIT) advancement [54,55]. In the healthcare sector, significant efforts have been directed towards promoting the adoption of EHRs due to their central role in health information management [13,56].

Key studies have yielded varied insights into EHR adoption. Kruse et al. [57] identified improvements in clinical document management and quality outcomes with EHR usage. Similarly, Sadoughi et al. [34,58] highlighted the significant impacts of perceived usability, intent, attitudes, performance, and social influence on EHR adoption.

The Technology Acceptance Model (TAM) and the Unified Theory of Acceptance and Use of Technology (UTAUT) are frequently employed in EHR adoption research. Kijsanayotin et al. [59] found that performance expectancy, effort expectancy, social influence, and attitudes were linked to the intention to use health IT in community health centers. Esmaeilzadeh et al. [60] reported that expected performance, self-efficacy, and social networks were critical factors among physicians regarding Clinical Decision Support Systems (CDSS) adoption. Additionally, Maillet et al. [61] indicated that performance expectancy was the most significant factor determining actual EHR use among nurses, followed by facilitating conditions impacting effort expectancy.

Almarzouqi et al. [62] integrated UTAUT and TAM to predict EHR system adoption in UAE healthcare, finding the intention to use EMRs as the most significant predictor of actual use. This study also highlighted the importance of factors like fear, innovation, self-efficacy, and trust in relation to TAM constructs.

The UTAUT model, which integrates elements from various models including TAM, TRA, TPB, and SCT, focuses on constructs such as effort expectancy, performance expectancy, facilitating conditions, and social influence. These constructs address the ease of learning and using a new system, the influence of coworkers, and how perceptions of support and resources impact use [63,64].

Our study expands on this body of research by exploring EHR adoption in the context of Jordanian nurses, focusing on their continuance intentions post-adoption. This approach aims to enhance understanding of sustained EHR use, filling a research gap in the application of UTAUT in healthcare settings, especially in the post-adoption phase [65].

## 2.2. The Big Five Model

The Big Five Model, also known as the Five-Factor Model, is the most widely accepted personality theory psychologists hold today. According to the Big Five model, personality comprises five basic components: openness to new experiences, neuroticism, extroversion, conscientiousness, and agreeableness [66]. Many more detailed features are offered within each of these five major components. The inclination to open up and explore new situations is referred to as openness to experience. Inquisitiveness and behavioural flexibility are examples of openness to experience [66,67]. Openness to experience, including components like creative thinking, aesthetic appeal, and intellect, describes a person's mental well-being's breadth, depth, and complexity [66,68]. Artistic and scholarly interests are related to openness to experience [69]. A sense of duty, responsibility and striving for achievement are part of integrity. Productivity, foresight, and self-control are qualities connected to conscientiousness [66,67]. Low levels of conscientiousness mean that impulse control is difficult, which can negatively affect planning and goal-oriented behaviour [66,68]. Open-minded social behaviour, spontaneity, and positive feelings characterise extraversion, and extraversion is associated with the urge for high activity, vitality, adventure-seeking, and social ability [66–68].

Extraversion is said to be linked with social and entrepreneurial interests [70]. People exhibiting lower levels of extraversion are called introverts [69]. Introverted people have a low level of extraversion [66]. Agreeableness is associated with trust and consideration for others, encompassing selflessness and kindness [66]. Being nice and considerate and avoiding disputes are all characteristics of agreeableness [66,67]. Agreeability contrasts with a hostile attitude toward people [66,68]. Neuroticism is related to negative emotions, particularly fear, and an inability to cope well with stress. Neurosis refers to symptoms of worry, tension, and negative effects [66,68]. Neuroticism contrasts with emotional stability, characterised by comfortable people who behave consistently independent of their surroundings [67]. Introverted people have a low level of extraversion [69]. Agreeableness is associated with trust and consideration for others, encompassing selflessness and kindness [69]. Being nice and considerate and avoiding disputes are all characteristics of agreeableness [66,67]. Agreeability contrasts with a hostile attitude toward people. Neuroticism is related to negative emotions, particularly fear, and an inability to cope well with stress. Neurosis refers to symptoms of worry, tension, and negative effects [66,68].

In the realm of incorporating personality variables into the Unified Theory of Acceptance and Use of Technology (UTAUT) models, a significant gap that Devaraj et al. [48] identified is the lack of emphasis on personality traits in existing research. This gap is due to the absence of a grounded theoretical approach for selecting specific personality traits relevant to UTAUT models.

Several pivotal studies have investigated the interplay of the Big Five personality model with Information and Communication Technology (ICT) adoption and use. Venkatesh et al. [71] discovered that extroversion, openness to new experiences, and contentiousness positively influence the adoption of e-government technology, highlighting the role of these traits in ICT usage. Lane and Manner [72] found a correlation between extroversion and the ownership and usage of smartphones, particularly for texting. In contrast, agreeable individuals are more inclined towards using smartphones for calling.

Wang [70] identified a relationship between extroversion, conscientiousness, and students' satisfaction with messaging apps, with these traits also influencing continued app usage. McElroy et al. [73] demonstrated the impact of openness to experience on Internet usage and found that neuroticism leads to an increased tendency for online shopping. Wilson et al. [74] conducted a survey among college students and found that personality traits and self-esteem are key predictors of social media usage and satisfaction levels.

Moore and McElroy [75] noted that personality traits, considering gender and Facebook experience, significantly influence patterns of social media behavior, including time spent on Facebook and types of activities engaged in. Svendsen et al. [76] integrated personality dimensions into the Technology Acceptance Model (TAM) and observed that traits like extraversion and conscientiousness impact user intentions through various TAM constructs.

Lin and Ong [77] modified TAM using five personality factors and a satisfaction variable, finding that personality traits indirectly affect ICT usage intentions, with openness to experience and agreeableness significantly influencing performance expectations. Zhou and Lu [78] explored smartphone commerce adoption, noting that neuroticism negatively impacts performance expectancy and trust, while agreeableness, extraversion, and openness to experience positively influence these factors.

Terzis et al. [79] developed a blended model for computer-based assessment acceptance among students, revealing how personality traits impact perceptions and intentions regarding ICT. Their findings highlighted that neuroticism negatively affects performance expectancy, while agreeableness and conscientiousness positively influence effort expectancy and social influence.

Alsyouf et al.'s [13] notable study integrated the Big Five Model with UTAUT to assess nurses' continuance intention to use electronic health records (EHRs). This research showed that personality traits significantly affect performance expectancy and continuance usage, with performance expectancy being moderated by conscientiousness in the relationship between social influence and continuance intention. The study identified a gap in understanding the role of organizational support and the Big Five personality traits through UTAUT variables on nurses' continuance intention to use EHRs. This gap indicates a need for further research to explore these relationships and their impact on technology adoption and usage in healthcare settings.

However, this body of research, while pioneering, also highlights a critical gap. It underscores the limited empirical exploration of nurses' roles in the post-adoption phase of EHRs, especially in developing countries like Jordan. Furthermore, there is a dearth of literature examining the combined impact of personality traits and organizational support on nurses' decision to continue using EHRs. This gap is significant as it overlooks how intrinsic individual factors and external organizational dynamics can jointly influence technology adoption and utilization in healthcare settings.

The present study proposes a comprehensive model that melds the Five-Factor Model (FFM) with UTAUT to address these limitations. This model aims to explore how personality traits and organizational support collectively shape nurses' decisions regarding the continued use of EHRs. By integrating these two frameworks, the study endeavours to provide a more holistic understanding of the antecedent role of personality traits and organizational support in influencing nurses' continuance intention with EHRs. This approach is expected to yield deeper insights into the nuanced interplay between individual characteristics and organizational factors, thereby contributing to a more informed strategy for technology implementation and utilization in healthcare settings.

## 2.3 Formulation of the proposed model and hypotheses

**2.3.1 Effort expectancy.**   Effort expectancy pertains to how simple a specific IS is to use [64]. The harder it is to implement an invention, the less likely customers will embrace it [80,81]. Effort expectancy can create barriers to the use of innovation. How expected effort is perceived depends on individual professional experience [64]. Having first-hand experience with new technologies shapes experiences. For example, an individual may anticipate that a certain website is easy to use. However, actual usage may confirm or contradict such a prediction. It has been demonstrated that effort expectancy affects the indirect impacts of usage intentions and attitudes [64]. This impact is probably true in continuity because people tend to act instrumentally without even realising it, regardless of their situation or stage [64]. Given that the human tendency to unconsciously seek active behaviours exists regardless of the occurrence or phase of such behaviours, this is probably true in continuation scenarios [82]. On the other hand, Alsyouf and Ishak [15] and Alsyouf et al. [12] discovered that effort expectancy positively impacted continuance intention. Consequently, the following hypothesis is posited:

H1. Effort expectancy will significantly affect the continuance intention to use EHRs.

**2.3.2 Performance expectancy.**   One description of performance expectancy is an individual's assumption about whether adopting an information technology system will likely increase their job productivity [64]. As a result, if caregivers have favourable expectations about using electronic health records, they are more likely to engage with them in the future. In his study, Bhattacherjee [82] discovered that perceptions of usefulness and anticipation for using the system increase performance and affect the desire for sustained use and initial acceptance of use. Zhou [80] employed UTAUT in his research to investigate the factors that influence the continuance of mobile Internet. The findings indicate that the performance expectancy positively impacted Continuance use. Alsyouf and Ishak [16] and Alsyouf et al. [13] found that performance expectancy positively influenced nurses' decision to continue their intention to use EHRs.

H2. Performance Expectancy will significantly affect the continuance intention to use EHRs.

**2.3.3 Social influence.**   Research into IS use recognises the importance of social influence in determining whether potential users agree that using an IS should be given particular consideration [83]. This factor has significantly impacted intention determination in many empirical trials [84]. It has been hypothesised that social influences can directly influence intentions. There is evidence that social influence can directly impact intention. Generally, the behaviour and opinions of others and any changes observed in friends and peers contribute to these perceptions of approval before usage. This factor has also been shown to positively influence CI in relevant studies [85,86]. Consequently, the following hypothesis is posited:

H3. Social influence will significantly affect the continuance intention to use EHRs.

**2.3.4 Facilitating conditions.**   A facilitating condition expresses one's belief regarding the existence of institutional and technological resources that enable them to use IS. [64]. A potential user's awareness is related to the person's capacity to manage the use of the information system. It remains widely assumed that this view will directly impact IS's intended use and application [64]. As long as they have sufficient resources to facilitate and enable the use of a

new IS, people can have a positive attitude towards it. A user's perceptions about technical assistance and information required will likely vary before and after using IS. Staff with greater access to resources when using IS, for example, access via devices and get help virtually through the Internet, allowing them to feel more comfortable using it. This positive affirmation translates to higher satisfaction levels and improves promotional conditions after use. These generate more positive attitudes and larger CIs to use [65].

Referring to the logic of the Expectation Confirmation Theory [38], positive confirmation of the facilitative condition positively affects the facilitative condition after use. Judging from existing studies, post-applied conditions positively affect CI and indirectly affect CI through post-applied settings [64]. Based on a literature review, Zhou [87] identified how facilitative conditions positively impact the use of continuations. On the other hand, Alsyouf and Ishaq [16] and Alsyouf et al. [13] noted that facilitative conditions positively influence nurses' intentions to use ongoing electronic health records. Consequently, the following hypothesis is posited:

H4. Facilitating conditions will significantly affect the continuance intention to use EHRs.

**2.3.5 Organisation support.** Previous studies have identified organisational support as one of the most important recurring factors influencing system success [13,16,52,88–90]. Organisational support can act as a change agent to ensure adequate allocation of resources and create an environment conducive to IS success [91–97]. Organisational support is therefore associated with greater system success, and its lack is seen as a significant barrier to the effective use of information technology.

Based on the literature on technology acceptance models, Davis et al. [98] suggested that beliefs (perceived usability and perceived usefulness) are influenced by organisational support. A high level of organisational support is thought to encourage more positive beliefs about the system in both users and caregivers [13,16,99]. Organisational support was found to be associated with positive beliefs and more system use. Moreover, lack of institutional support is seen as a significant obstacle to the effective use of electronic health records [13,16,99]. Organisational support has been found to be directly related to EHR adoption [13,16,99].

According to Igbaria [111] and Igbaria and Chakrabarti [112], organisational support combines two broad categories. Providing system development assistance, specific instructions and assistance regarding the use of electronic health records; Management support such as resource allocation and encouragement from top management. Consequently, the following hypotheses are posited.

H5. Facilitating conditions will be positively affected by end-user support.

H6. Performance Expectancy will be positively affected by management support.

*2.3.5.1. Neuroticism.* Nurses who value neuroticism are more prone to unpleasant feelings such as fear, worry, sorrow, humiliation, anger, melancholy, hostility, and guilt. They frequently have unreasonable notions and have a harder time dealing with stress. Furthermore, they are prone to unpleasant feelings and reactions. People with hypo neurosis may handle unpleasant conditions with minimal emotion. [100]. Neurotic nurses endure more unfavourable situations than other nurses due to their negative disposition. Therefore, these nurses are expected to struggle in recruitment whenever new issues arise, for instance, the use of EHRs. Neurotic nurses perceive technological advances as stressful and threatening and do not expect them to help them perform their duties. They have been reported to exhibit adverse emotions concerning ICT that they are unfamiliar with. [71]. As a result, a neurotic nurse realises that

they must exert considerable effort to utilise an EHR if they lack organisational and technological assistance. Consequently, the following hypothesis is posited:

H7. Facilitating conditions are adversely affected by neuroticism.

*2.3.5.2. Extraversion.* Extroverted nurses tend to be goal-oriented, gregarious, confident, energetic, and conversational. They love engaging with others and can keep interpersonal interactions going. They like a challenge and are prone to experiencing pleasant feelings. Extremely introverted people are reserved, task-oriented, and silent. [100]. Extroverted nurses tend to focus on developing relationships with others and considering the image they portray. In addition, Venkatesh and Davis [84] found that if most members of society hold an opinion which suggests extroverted caregivers need to utilise ICT, they should use ICT to develop and sustain a positive image in the community. Furthermore, Wang and Yang [101] and Lin [102] examined the relationship between personality traits and e-learning technology acceptance. The results show that extraversion positively correlates to the easy use of e-learning technology. Zhang Zhang [103] also investigated the relationship between personality traits and acceptance of social networking sites. They found that extraversion positively correlates with the ease of using social networking sites. Overall, these studies suggest that people with high extroversion find it less taxing or easier to use new technology than people with low extroversion. Consequently, the following hypotheses are posited:

H8. Social influence is positively affected by extraversion.

H9. Effort expectancy is positively affected by extraversion.

*2.3.5.3. Openness to experience.* Art and aesthetics are important to caregivers who appreciate openness. Caregivers who appreciate diverse perspectives possess a strong intellectual curiosity and can enjoy new experiences, deal with anxieties, and go on adventures. They have unusual ideas, ideals, and beliefs. Nurses who are not receptive to new experiences are traditional and narrow-minded. [100]. Caregivers open to new experiences will be interested in trying the EHR and appreciating its application in the context of technological adoption and utilisation. Sanjebad and Iahad [104] proposed that perceived usefulness would be positively affected by openness to experience, corresponding toward fulfilment expectations under the UTAUT theory. According to these scholars, nurses more open to experience are more likely to seek new possibilities to demonstrate their innovative skills. This attribute is crucial when incorporating valuable modern medical technologies like EHRs. Zhou and Lu [78] found a positive relationship between openness to experience and performance expectancy, mediated by trust. Devaraj et al. [48] and Lin and Ong [77] supported this result, reporting a positive direct effect of openness to experience on performance expectancy. This strongly indicates that openness to experience is an important factor in performance expectancy. Thus, organisations must create an environment encouraging openness and trust, leading to more empowered and productive employees.

Additionally, technical systems often require active learning, so being open helps identify users eager to learn and use technology in the classroom. In a study of student Internet use, McElroy et al. [73] found openness to experience to predict overall use. Agarwal and Karahanna [105] investigated the link between cognitive absorption and the utility and usability of IT systems in a related study. The authors defined cognitive absorption as a "response to inductive stimuli" (p. 667) in which a person gets intensely interested and involved in a particular activity. Theoretically, cognitive immersion coincides with openness, characterised as "inquiring intelligence" ([106] p. 423). They also showed that cognitive absorption predicts utility, usability, and intention to use IT systems. Consequently, the following hypotheses are posited:

H10. Performance expectancy is positively influenced by openness to experience.

H11. Effort expectancy is positively influenced by openness to experience.

*2.3.5.4. Agreeableness.* Altruistic nurses are friendly and agreeable, collaboration is one of their favourite activities, and they are always willing to assist others. They are tolerant and care deeply about social peace. They are more motivated to attain interpersonal connections, which should result in increased happiness. It is competitive rather than cooperative [100]. Agreeable nurses demonstrate strong adherence to protocol and trustworthiness with colleagues, organisations, and authorities. Thus, they will find it simple to use an EHR since it will provide them with an organisational and technical framework whenever required. Most are extremely loyal to authority and will use EHR if management recommends it. Moreover, the opinions of others are also shown to influence the behaviour of agreeable nurses [48,79]. As a result, agreeableness can have a favourable influence on social influence. Furthermore, McElroy et al. [73] asserted that a tolerant attitude among well-motivated nurses was likely to enable nurses to be more committed and patient even when confronted with unsatisfactory Information Systems usage, making them more likely to attempt for a longer period than other nurses. Consequently, the following hypotheses are posited:

H12. Effort expectancy is positively influenced by agreeableness.

H13. Facilitating conditions are positively affected by agreeableness.

H14. Social influence is positively influenced by agreeableness.

*2.3.5.5. Conscientiousness.* Conscientious nurses are neat, disciplined, and motivated. They are well-known for excelling at their occupations. They are patient nurses who take measures to enhance their performance. They are responsible, timely, and dependable. People with poor conscientiousness are untrustworthy, indolent, thoughtless, weak-willed, and hedonistic [100]. Sanjebad and Iahad [104] proposed that conscientiousness positively affects perceived usefulness, which relates to the UTAUT model's performance expectancy. Highly conscientious nurses recognise how EHRs are practical, improve job performance, and are useful, leading to acceptance of usage. Consequently, the following hypothesis is posited.

H15. Performance expectancy is positively influenced by conscientiousness.

## 3. Methodology

### 3.1 Design, sample and data collection

This study used a quantitative approach and cross-sectional design to test the proposed model and systematic random sampling, a probability sampling method. The target population was nurses with knowledge and access to EHRs employed in Jordanian hospitals. An initial sample frame was derived from an email list representing public hospitals with fully functional EHRs, and in the second stage, the sample frame was derived from a list of nurses who worked in those hospitals. The survey included ten hospitals. Each hospital department head submitted a list of nurses employed at their facility, with 2194 nurses distributed across the ten hospitals sample.

Consistent with Krejcie and Morgan [107], an appropriate sample size should range between 322 and 327 in a study of this size. To determine the number of responses required from each hospital, we used this mathematical formula: "(number of nurses in hospital x/total number of nurses in all hospitals) × sample size." The researchers collected a list of nurses' names from their department heads. After that, one name was randomly chosen after every

5th name in the list. The researchers distributed 510 questionnaires in person using systematic random sampling to guarantee a minimum of 327 responses. Of the 510 surveys distributed, 490 were returned; 18 were eliminated because they lacked half of the necessary information [108]. Therefore, this study used 472 surveys, producing an effective response rate of about 92.5%.

Ethics approval and consent to participate Ethical approval to conduct the study was obtained from the Jordanian Ministry of Health (MOH) and the participating hospitals with reference number (MBA / Ethics Committee / 10908).

## 3.2. The measurement of constructs

This study used constructs and items tested in other studies to assure discriminant and convergent validity. Six items from [13,16,65] were used to represent CI. In addition, five items for performance expectancy, five for effort expectancy, eight for social influence, and six for facilitating conditions were adopted from [13,16,65,109]. Following Igbaria [110] and Igbaria and Chakrabarti [111], eight items of organisational support were combined as an indicator of two broad categories. The scale consisted of eight components, four representing end-user assistance and four representing managerial support. This study employed The NEO Five-Factor Inventory (NEO-FFI) that Costa and McCrae developed [112]. It consists of a 60-item scale, with 12 items dedicated to each of the five dimensions—neuroticism, extraversion, openness to experience, agreeableness, and conscientiousness. These dimensions are robustly validated and have been extensively used in prior studies [13,48,113,114] to measure the Big Five personality traits.

Brislin's [115] guidance in translating the questionnaire from English to Arabic was adopted. Consequently, four academics with extensive IS expertise assessed the questionnaire before the study. Following the pre-test, the questionnaire was subjected to a few minor changes. All the survey responses were assessed on a 5-point Likert scale (1 = strongly disagree, 5 = strongly agree).

## 4. Results

Table 1 shows that 30.1% of the respondents were male and 69.9% were female. Most respondents, 58.5%, were between 26 and 35 years old. Most respondents had a bachelor's degree (76.7%). The respondents' work experience distribution showed that 53.2% had less than ten years of work experience. A wide range of departments were covered in this study, including ICU, CCU, ER, dialysis, ward, and operating room, with 9.5%, 4.7%, 9.7%, 9.3%, 47.2%, and 19.5%, respectively. See Table 1.

Structural Equation Modelling (SEM) in SmartPLS 4 software analysed the research model and the fifteen hypotheses. The study investigated the causal models, including the measurement and structural models. Confirmatory factor analysis (CFA) estimated the measurement model to determine if the constructs had appropriate reliability and validity. The structural model investigated the direction and degree of relationships between the postulated components.

Segars and Grover [116] suggested assessing the measurement model first and then re-specifying it to attain the best model fit. The model's initial inspection revealed that several components needed to be eliminated. The instrument re-specification resulted in the retention of 54 elements.

Cronbach's alpha (α) for all construct more than 0.60 [117,118], The indicators' reliability was determined by item loadings greater than 0.60 for all items [119]. The average value of all

**Table 1. Demographics for the sample.**

| Items | Category | Frequency | % |
|---|---|---|---|
| Gender | Male | 142 | 30.1 |
| | Female | 330 | 69.9 |
| | Total | 472 | 100 |
| Age | 20–25 | 53 | 11.2 |
| | 26–30 | 153 | 32.4 |
| | 31–35 | 123 | 26.1 |
| | 36–40 | 67 | 14.2 |
| | 41–45 | 58 | 12.3 |
| | 46–50 | 15 | 3.2 |
| | 51–55 | 3 | 0.6 |
| | Total | 472 | 100 |
| The level of education | Diploma | 80 | 16.9 |
| | Bachelor's degree | 362 | 76.7 |
| | Master's Degree | 28 | 5.9 |
| | PhD | 2 | 0.4 |
| | Total | 472 | 100 |
| Profession Experience | 1–5 yrs. | 128 | 27.1 |
| | 6–10 yrs. | 123 | 26.1 |
| | 11–15 yrs. | 104 | 22 |
| | 16–20 yrs. | 74 | 15.7 |
| | 21–25 yrs. | 37 | 7.8 |
| | 26–30 yrs. | 5 | 1.1 |
| | 31–35 yrs. | 1 | 0.2 |
| | Total | 472 | 100 |
| Hospital Name | Princess Rahma | 60 | 12.7 |
| | Princess Badeea | 22 | 4.7 |
| | Prince Hamzah | 87 | 18.4 |
| | Prince Hussein Ben Abdulla II | 36 | 7.6 |
| | Al Zarqa | 96 | 20.3 |
| | Al Mafraq | 29 | 6.1 |
| | Maternity and Children—Mafraq | 39 | 8.3 |
| | Al-Ramtha | 23 | 4.9 |
| | Ma'an | 42 | 8.9 |
| | Queen Rania | 38 | 8.1 |
| | Total | 472 | 100 |
| Department Name | ICU | 45 | 9.5 |
| | CCU | 22 | 4.7 |
| | ER | 46 | 9.7 |
| | Dialysis | 44 | 9.3 |
| | Ward | 223 | 47.2 |
| | Operation Theatre | 92 | 19.5 |
| | Total | 472 | 100 |

constructions was greater than 0.5. The composite reliability varied from 0.783 to 0.947, corresponding to the acceptable value of 0.70 recommended Hair, Sarstedt [120]. See Table 2.

AVE square roots were calculated for each construct to determine discriminant validity; all square roots were significantly greater than correlations between constructs, demonstrating

**Table 2. Measurement model: Convergent validity.**

| Constructs | Measurement Items | Loadings | Cronbach's Alpha | Composite reliability (rho_c)[b] | AVE[a] |
|---|---|---|---|---|---|
| Effort Expectancy | EE1 | 0.842 | 0.884 | 0.916 | 0.688 |
| | EE2 | 0.883 | | | |
| | EE3 | 0.871 | | | |
| | EE4 | 0.67 | | | |
| | EE5 | 0.862 | | | |
| Performance Expectancy | PE1 | 0.895 | 0.877 | 0.924 | 0.803 |
| | PE2 | 0.918 | | | |
| | PE3 | 0.874 | | | |
| Facilitating Conditions | FC1 | 0.82 | 0.868 | 0.905 | 0.656 |
| | FC2 | 0.852 | | | |
| | FC3 | 0.797 | | | |
| | FC4 | 0.816 | | | |
| | FC5 | 0.761 | | | |
| End User Support | EUS1 | 0.81 | 0.79 | 0.864 | 0.615 |
| | EUS2 | 0.803 | | | |
| | EUS3 | 0.809 | | | |
| | EUS4 | 0.709 | | | |
| Management Support | MS1 | 0.796 | 0.787 | 0.858 | 0.602 |
| | MS2 | 0.785 | | | |
| | MS3 | 0.806 | | | |
| | MS4 | 0.713 | | | |
| Neuroticism | N11 | 0.884 | 0.762 | 0.843 | 0.576 |
| | N2 | 0.682 | | | |
| | N6 | 0.748 | | | |
| | N9 | 0.705 | | | |
| Extraversion | E11 | 0.791 | 0.697 | 0.815 | 0.527 |
| | E4 | 0.612 | | | |
| | E7 | 0.698 | | | |
| | E8 | 0.789 | | | |
| Openness to Experience | O2 | 0.832 | 0.602 | 0.783 | 0.55 |
| | O3 | 0.771 | | | |
| | O8 | 0.603 | | | |
| Agreeableness | A1 | 0.799 | 0.714 | 0.823 | 0.539 |
| | A10 | 0.763 | | | |
| | A4 | 0.716 | | | |
| | A7 | 0.65 | | | |
| Conscientiousness | C12 | 0.733 | 0.729 | 0.83 | 0.551 |
| | C2 | 0.698 | | | |
| | C5 | 0.766 | | | |
| | C8 | 0.769 | | | |
| Social Influence | SI1 | 0.632 | 0.882 | 0.907 | 0.554 |
| | SI2 | 0.714 | | | |
| | SI3 | 0.835 | | | |
| | SI4 | 0.86 | | | |
| | SI5 | 0.813 | | | |
| | SI6 | 0.767 | | | |
| | SI7 | 0.688 | | | |

*(Continued)*

**Table 2.** (Continued)

| Constructs | Measurement Items | Loadings | Cronbach's Alpha | Composite reliability (rho_c)[b] | AVE[a] |
|---|---|---|---|---|---|
| | SI8 | 0.603 | | | |
| Continuance Intention | CI1 | 0.853 | 0.933 | 0.947 | 0.75 |
| | CI2 | 0.845 | | | |
| | CI3 | 0.829 | | | |
| | CI4 | 0.893 | | | |
| | CI5 | 0.904 | | | |
| | CI6 | 0.869 | | | |

Note

[a] Average Variance Extracted (AVE) = (summation of the square of the factor loadings)/ [(summation of the square of the factor loadings) + (summation of the error variances)]

[b] Composite Reliability (CR) = (square of the summation of the factor loadings)/[(square of the summation of the factor loadings) + (square of the summation of the error variances)].

discriminant validity (Table 3). This finding resulted in a satisfactory evaluation of the measurement model, including convergent and discriminant validity measures. Following Cohen [121], endogenous latent variables with $R^2$ values of 0.26 are defined as substantial), 0.13 as moderate, and 0.02 as weak. The $R^2$ value of the entire model was 0.343. Moreover, the $R^2$ scores related to the variables of Continuance intention, facilitating conditions, and performance expectancy appeared to be very strong, and the $R^2$ values for effort expectancy and social influence have been moderate. The $R^2$ score was 0.665 regarding Continuance intention, revealing that the facilitating conditions, effort expectancy, social influence, and performance expectancy constructs explained 66.5% of the variance in the Continuance intention of Jordanian nurses to use electronic health records. See Table 3.

Once the measurement model is analysed, the next step in a PLS analysis is to evaluate the structural models to test the theoretical hypotheses. The hypothesis tests are based on the structural model's test results. A bootstrapping procedure with 5000 random resamples, and 472 cases per sample was conducted sequentially to estimate the path coefficients between the

**Table 3. Measurement model: Discriminant validity.**

| | AVE | A | C | CI | E | EE | EUS | FC | MS | N | O | PE | SI |
|---|---|---|---|---|---|---|---|---|---|---|---|---|---|
| A | 0.539 | 0.734 | | | | | | | | | | | |
| C | 0.551 | 0.706 | 0.742 | | | | | | | | | | |
| CI | 0.75 | 0.401 | 0.4 | 0.866 | | | | | | | | | |
| E | 0.527 | 0.645 | 0.679 | 0.364 | 0.726 | | | | | | | | |
| EE | 0.688 | 0.462 | 0.437 | 0.699 | 0.417 | 0.829 | | | | | | | |
| EUS | 0.615 | 0.33 | 0.248 | 0.353 | 0.253 | 0.42 | 0.784 | | | | | | |
| FC | 0.656 | 0.362 | 0.309 | 0.49 | 0.28 | 0.529 | 0.583 | 0.81 | | | | | |
| MS | 0.602 | 0.265 | 0.186 | 0.406 | 0.222 | 0.448 | 0.746 | 0.611 | 0.776 | | | | |
| N | 0.576 | 0.28 | 0.299 | 0.061 | 0.21 | 0.107 | -0.05 | 0.094 | -0.07 | 0.759 | | | |
| O | 0.55 | 0.678 | 0.681 | 0.343 | 0.575 | 0.408 | 0.252 | 0.286 | 0.227 | 0.226 | 0.742 | | |
| PE | 0.803 | 0.355 | 0.363 | 0.757 | 0.321 | 0.62 | 0.379 | 0.455 | 0.462 | 0.052 | 0.313 | 0.896 | |
| SI | 0.554 | 0.382 | 0.349 | 0.523 | 0.307 | 0.532 | 0.526 | 0.496 | 0.569 | -0.06 | 0.331 | 0.533 | 0.744 |

A = agreeableness, C = conscientiousness, CI = Continuance intention, E = extraversion, EE = effort expectancy, EUS = end user support, FC = facilitating conditions, MS = management support, N = neuroticism, O = openness to experience, PE = performance expectancy, and SI = social influence.

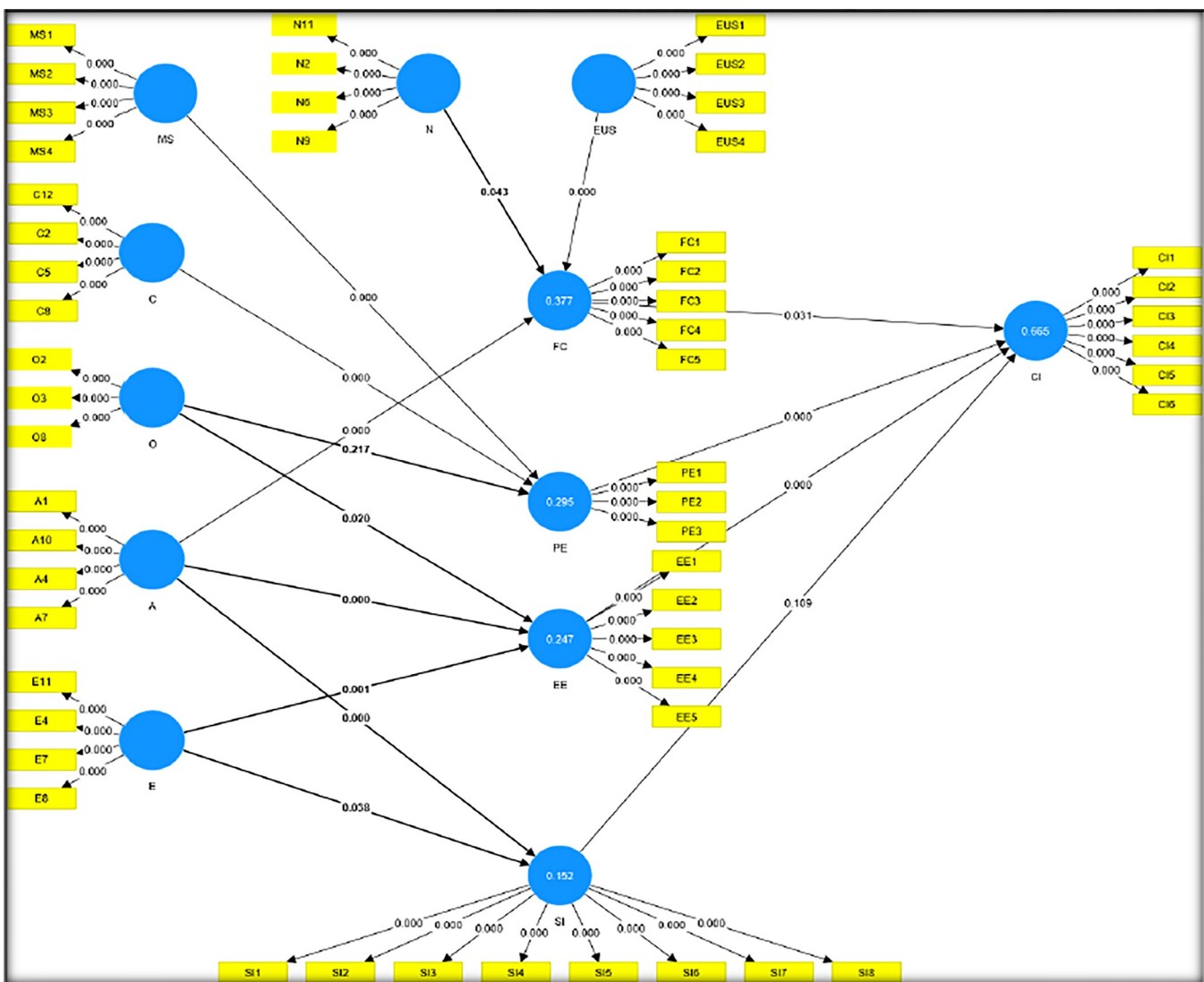

**Fig 1. Path coefficients assessment and hypotheses testing.**

constructs. A T-statistic was calculated to assess the significance of the path coefficients. Some relationships, however, were insignificant. Based on the results of the PLS structural model assessment, Fig 1 presents the overall explanatory power of the PLS model as well as the associated p-values of each significant path as indicated below: t values > 1.645 (p < 0.05); t values > 2.33 (p < 0.01); t values > 3.09 (p < 0.001)1-tailed test [122].

Using the bootstrap resampling procedure, significance tests were conducted on all paths. Table 4 shows the results.

A = agreeableness, C = conscientiousness, CI = Continuance intention, E = extraversion, EE = effort expectancy, EUS = end user support, FC = facilitating conditions, MS = management support, N = neuroticism, O = openness to experience, PE = performance expectancy, and SI = social influence.

Based on the hypotheses of this study, Table 4 shows that the results indicated that effort expectancy (β = 0.331, p < 0.001), performance expectancy (β = 0.496, p < 0.001), and facilitating conditions (β = 0.064, p < 0.05) significantly influenced nurses' intention to continue

**Table 4. Testing hypothesis and path coefficients for the structural model.**

| Hypothesis | Direction | Beta | Std Error | t-value | P values | Decision |
|---|---|---|---|---|---|---|
| H 1 | EE -> CI | 0.331 | 0.056 | 5.873*** | 0.000 | supported |
| H 2 | PE -> CI | 0.496 | 0.046 | 10.733*** | 0.000 | supported |
| H 3 | SI -> CI | 0.051 | 0.042 | 1.229 | 0.109 | Not |
| H 4 | FC -> CI | 0.064 | 0.034 | 1.861* | 0.031 | supported |
| H 5 | EUS -> FC | 0.531 | 0.038 | 13.936*** | 0.000 | supported |
| H 6 | MS -> PE | 0.404 | 0.041 | 9.969*** | 0.000 | supported |
| H 7 | N -> FC | 0.071 | 0.042 | 1.713* | 0.043 | supported |
| H 8 | E -> SI | 0.103 | 0.058 | 1.771* | 0.038 | supported |
| H 9 | E -> EE | 0.172 | 0.058 | 2.981** | 0.002 | supported |
| H 10 | O -> PE | 0.047 | 0.06 | 0.781 | 0.217 | Not |
| H 11 | O -> EE | 0.131 | 0.063 | 2.061* | 0.020 | supported |
| H 12 | A -> EE | 0.263 | 0.071 | 3.678*** | 0.000 | supported |
| H 13 | A -> FC | 0.167 | 0.043 | 3.899*** | 0.000 | supported |
| H 14 | A -> SI | 0.315 | 0.057 | 5.523*** | 0.000 | supported |
| H 15 | C -> PE | 0.256 | 0.055 | 4.689*** | 0.000 | supported |

Notes: t values are calculated through bootstrapping routine with 473 cases and 5000 samples.

*$P < 0.05$

** $P < 0.01$

***$P < 0.001$ (One-Tailed test).

using electronic health records, supporting hypotheses $H_1$, $H_2$, and $H_4$. However, social influence ($\beta = 0.064$, $p < 0.05$) did not significantly influence nurses' intention to continue using EHRs, so H3 was rejected. The influence of end-user support ($\beta = 0.531$, $p < 0.001$) positively and significantly affected the facilitating conditions, thus supporting ($H_5$). As for management support, the study found that management support positively influenced performance expectancy ($b = 0.404$, $p$ 0.001); thus, $H_6$ was confirmed. The results suggest that neurotic nurses ($\beta = 0.071$, $p < 0.05$) negatively influenced facilitating conditions, supporting ($H_7$). In addition, extraversion positively affected social influence ($\beta = 0.103$, $p < 0.05$) and effort expectancy ($\beta = 0.172$, $p < 0.01$), supporting ($H_8$, $H_9$). In addition, a positive relationship was also found between openness to experience and effort expectancy ($\beta = 0.047$, $p > 0.05$) and showed no effect on performance expectancy ($\beta = 0.131$, $p < 0.05$). Consequently, $H_{10}$ was rejected, and $H_{11}$ was supported. The study findings showed that more agreeable nurses were more willing to expect effort ($\beta = 0.263$, $p < 0.001$), facilitate conditions ($\beta = 0.167$, $p < 0.001$), and exert social influence ($\beta = 0.315$, $p < 0.001$) as a result of their interpersonal skills and cooperative nature, which supported ($H_{12}$, $H_{13}$, $H_{14}$). Finally, conscientiousness ($\beta = 0.256$, $p < 0.001$) positively influenced performance expectancy; therefore, $H_{15}$ was confirmed.

## 5. Discussion

An analysis of the data indicated a positive and significant association existed between effort expectancy and intention to continue using EHRs (H1). A similar finding is supported by the UTAUT, which asserts that effort expectancy directly affects technology adoption and CI [13,16,123]. Nurses spend their time treating patients, administering medications, and performing laboratory tests. The effort required to use the EHR influences the nurse's use and intent to use her EHR. Therefore, a user-friendly interface for the EHR is an important topic to focus on when developing the EHR.

Furthermore, the findings support the relevance of performance expectancy on nurses' Continuance intention to utilise the technology (H$_2$). Performance expectancy was strongly linked with enthusiasm for use, consistent with UTAUT principles [13,16,65], which argue that greater performance expectancy among nurses boosts their willingness to utilise electronic health records. Due to this, nurses are interested in learning how EHRs can help them provide more efficient and effective care.

Social influence can be a double-edged sword regarding technology adoption (H$_3$). The results show that Social influence did not significantly affect nurses' intentions to continue using EHR, consistent with [16]. On the one hand, social influence can encourage individuals to adopt technology by providing social validation, support, and pressure. Social influence can also discourage users from adopting the technology, especially if there is resistance or scepticism in their social networks.

The link between social influence and nurses' inclination to adopt EHRs is complicated and situation dependant. Social influence perceptions may be adjusted by nurses based on their observations of others' performance, new information, and/or changes in the opinions of their peers. Upon using an EHR, nurses will develop their own perceptions and convictions. It is based on their productivity, performance, and ease of use. Nurses' decisions to adopt and use the new system become less dependent on the opinions of others after it has been used [16].

According to the findings, facilitating conditions such as accessibility and easy availability of essential resources and expertise benefit nurses' desire to continue using EHRs after initial usage (H4), consistent with previous studies [16,65,87]. When facilitating conditions support nurses' needs, they are more likely to continue to use electronic health records (EHRs). Access to essential resources and expertise is one of these conditions. Hospitals that provide nurses with the necessary resources can support them using EHRs. As a result of the training and resources they receive, they are better able to access centralized information, including data submitted by other clinicians. In addition to streamlining nurses' workflow, this easy access to information also creates enthusiasm for using the system among nurses. With adequate resources, nurses develop a positive attitude toward using the system. As they utilize the EHR, they evaluate the appropriateness of the resources (such as relevant knowledge and assistance). Nurses believe that the facilitating conditions are favorable if the resources exceed their expectations. As a result of this positive perception, a user intends to continue using the system. Therefore, nurses are more likely to remain committed to EHRs when they have the appropriate support. Organisational support is a combination of two broad categories, according to Igbaria [110] and Igbaria and Chakrabarti [111] (H$_5$, H$_6$). End-user support may have a positive impact on facilitating conditions for nurses by providing them with the resources, training, and support that they require to maintain high standards of care. Health professionals can benefit from end-user support by gaining access to the equipment, supplies, and technology they need to provide high-quality care. Additionally, end-user support can provide technical assistance to health professionals so that they can solve problems and overcome technical barriers they may encounter during patient care. A supportive environment conducive to quality care can be created by providing end-user support to healthcare professionals.

To foster an environment in which healthcare workers are encouraged to adopt new technologies and practices, management support can positively impact nurses' performance expectancies. For example, managerial support can effectively communicate to nurses the benefits and goals of new technologies or practices. In addition, management support can help to allocate resources effectively and ensure that nurses have the tools and support they need to utilize new technologies or practices. Nursing managers can facilitate the adoption of new technologies and practices by providing nurses with this support. As a result, a lack of organizational support is considered a major barrier to the effective use of EHRs [13,16,99]. Thus, there is

evidence that organizational support is directly related to the adoption of electronic health records [13,16,99].

Regarding hypotheses 5 and 6, the combination of end-user support (providing training, assistance, and instructions) and management support (resource allocation, encouragement, and involvement) creates an environment that positively influences nurses' experience with the EHR system. The support enhances the facilitating conditions for nurses, making the system more accessible and user-friendly. Moreover, it boosts nurses' performance expectancy by increasing their belief in the system's usefulness, motivating them to utilize the EHR effectively for improved patient care.

The findings imply that neurotic nurses negatively impact facilitative conditions (H7). Neurotic nurses are more likely to experience negative occurrences than other nurses, in part because they put themselves in situations that encourage adverse effects [124]. Neurotic nurses may negatively influence healthcare working conditions by creating work environments that are stressful and challenging for themselves and their colleagues. As a result, when confronted with new concerns such as EHR use, these nurses are likely to struggle with adoption. It has been found that neurotic nurses resist change and have unfavourable views regarding ICTs that they have not yet encountered [71], which would be the case with EHRs. Neurotic nurses believe that implementing EHRs requires enormous work and, without organisational support, leads to lower levels of innovation and productivity, which can impair patient care quality.

As a result, when neurotic nurses perceive a lack of organizational and technological assistance in utilizing EHRs, they realize they must exert considerable effort to overcome these challenges. The negative influence of neuroticism on facilitating conditions suggests that these nurses face difficulties accessing the necessary support and resources to use EHRs effectively. The result highlights the importance of considering the psychological disposition of nurses, particularly neuroticism when implementing and supporting EHR systems. By understanding the potential challenges faced by neurotic nurses, healthcare organizations can provide targeted support and assistance to mitigate the adverse effects of neuroticism on facilitating conditions and promote successful EHR adoption and usage.

This study found that extraversion positively impacted social influence and effort expectancy (H$_8$, H$_9$). Venkatesh and Davis [84] found that an extraverted nurse is motivated to use ICT when influential people think they should engage in this activity to sustain a positive image in society.

Extroverted nurses, who are characterized as goal-oriented, gregarious, confident, energetic, and conversational, tend to focus on developing relationships with others and considering the image they portray. When it comes to utilizing information and communication technology (ICT), if most members of society believe that extroverted caregivers should use ICT, they are more likely to do so to develop and sustain a positive image in the community. This positive perception of extroverted nurses using technology can impact social influence positively.

Also, extraversion has a positive effect on effort expectancy. This means extroverted individuals find it less taxing or easier to use new technology than those with low extroversion. Previous studies examining personality traits and technology acceptance, such as e-learning technology and social networking sites, suggest that people with high extroversion tend to use new technology easily. Extroverted nurses who enjoy engaging with others are more likely to embrace technological tools and perceive them as less challenging. This positive attitude towards technology contributes to higher effort expectancy.

The results demonstrate that extraversion positively influences both social influence and effort expectancy. With their outgoing and confident nature, extroverted nurses tend to be

more influenced by social perceptions and find it easier to adopt and use new technology. Healthcare organizations can leverage these characteristics to promote successful technology implementation and utilization among nurses by understanding the role of extraversion in technology acceptance.

This study found that openness to experience positively affected effort expectancy but not performance expectancy ($H_{10}$, $H_{11}$). Effort expectancy is the degree of ease a person associates with using a particular technology or practice. Through openness to new experiences, problem-solving skills, and a positive attitude, nurses can create a culture of learning and innovation that positively impacts effort expectancy. This culture can make it easier for health professionals to adopt and use new technologies or practices, ultimately improving patient outcomes.

On the other hand, the link between openness to new experiences and performance expectancy can be complex and depends on several circumstances. The intrinsic aspects of openness to experience on performance expectancy are poorly understood, partly due to a lack of appropriate study in the scientific literature [125]. Researchers have concluded that the link between openness to new experiences and performance expectations is inconclusive and may not be significant [125,126]. Barrick, Parks [127] referred to openness to experience as a "contingent predictor" (p. 748) because its importance to job performance likely depends on job demands. Raja and Johns [128] discovered that task context influences the effect of openness on creativity, which might be regarded as a component of discrete task context in Johns [129] framework. In addition, Neal, Yeo [130] found that individual openness is negatively related to teamwork and organisational skills, which the authors believe may limit cooperative behaviours in the workplace. Outgoing individuals are more likely to seek out challenges and learning opportunities, which might improve their ability and confidence to perform certain tasks. These individuals also use proactive problem-solving strategies, which could also lead to higher performance expectancy. Nurses' openness to experience in our study did not significantly affect nursing performance expectancy.

One possible reason that openness to experience can contribute to a positive attitude towards technology adoption is that performance expectancy is a multifaceted construct shaped by various factors, including training, system usability, and organizational support. Openness to experience alone may not be sufficient to influence performance expectancy significantly.

Moreover, the results suggest that more agreeable nurses are more willing to expect effort, facilitate conditions, and exert social influence because of their interpersonal skills and cooperative nature ($H_{12}$, $H_{13}$, $H_{14}$). Ultimately, their positive attitudes and willingness to collaborate can contribute to the creation of a culture of learning and innovation that can ultimately lead to improved patient outcomes. According to this study, nurses who are agreeable are more likely to collaborate and assist, which can facilitate the adoption of new technologies or practices, as well as lead to a higher level of expectation of effort. Furthermore, nurses who are agreeable are likely to have positive attitudes toward helping their colleagues and patients, which may increase their confidence and perception of comfort in using technology or practices. Furthermore, nurses who are more agreeable are more likely to be cooperative, supportive, and helpful to their colleagues, which can enhance the working environment [48,79]. It is possible to create a positive environment by working with others in order to overcome technical or organizational barriers. Furthermore, nurses who are agreeable tend to be more effective communicators and better at building relationships, which can positively impact their social influence. As a result of their interpersonal skills, they can influence their colleagues to adopt and use the EHR technologies, and they can serve as positive role models for their colleagues, which will increase their perceived ease and confidence with the use of EHRs in hospitals.

Conscientiousness was found to positively influence performance expectancy ($H_{15}$) in this study. It has been shown that nurses with a high level of conscientiousness have a positive effect on performance expectancy due to their responsibility, hard work, and reliability [104]. A conscientious nurse assumes responsibility and ensures that all goals and objectives are met. They are often diligent in their work and have a strong desire to achieve their goals, which can lead to higher performance expectations. This desire can help them identify potential problems or errors when using EHRs, increasing their confidence in using these technologies effectively. Increased confidence can lead to higher performance expectancy among their colleagues, helping them prioritise their tasks and focus on using EHRs to achieve their goals, leading to higher performance expectations.

## 6. Conclusion

In this study, the five-factor model and UTAUT were used. The role of personality in individual responses to technology, specifically nurses' intentions to continuously use an EHR system, was examined. The UTAUT model is described in this work, emphasising the antecedent role of human personality and organisational support in nurses' EHR continuous use from the UTAUT perspective. Statistically, the findings indicate a significant association between Effort Expectancy, performance expectation, and facilitating conditions, and a willingness to continue using electronic health records. The association relating social influence and Continuance Intention to Use Electronic Health Record Systems, on the other hand, did not appear substantial. Social influence may have little effect on nurses' intentions to utilise EHRs. The study also found that end-user support tremendously affects facilitating conditions. As for management support, the study found that management support directly affects performance expectancy. The results also suggest that neurotic nurses negatively influence supportive conditions. Accordingly, these nurses can be expected to have problems with acceptance when confronted with new issues such as EHR use. In addition, extraversion positively affected and supported social influence and effort expectancy. In addition, openness to experience positively affected effort expectancy and showed no effect on performance expectancy. Results showed that more agreeable nurses were more likely to expect effort, facilitate conditions, and exert social influence because of their interpersonal skills and cooperative nature. Finally, conscientiousness positively influenced performance expectancy.

### 6.1 Practical implications

Several key practical implications are presented in this study for those in charge of the healthcare industry. These groups include healthcare institution managers, government agencies, health information system consultants and vendors, and hospitals. The outcomes are relevant to introducing technology into the healthcare industry at the individual and organisational levels. A process such as this has the potential to improve healthcare systems.

A HIS may not be adopted and used efficiently if the organisation does not encourage and assist employees to use it continuously. If employees are not provided with assistance using the system to complete or complement their work, use will likely cease. Managers responsible for implementing IS innovations in designing a well-organised strategy face a significant challenge regarding the importance of managerial attitude toward EHR implementation and employee behaviour. Actions speak louder than words: how top managers behave, particularly in their interactions with staff, directly impacts the success of EHR adoption. A manager's practical, active, and relational behavior transforms employee behaviour and fosters the sharing of vision among the team members. Top managers should constantly demonstrate support and be readily available to assist staff in integrating management's strategic vision with their individual performance

goals. Employees can share observations and experiences and actively respond to issues or questions. For nurses to successfully use EHRs, top management support is considered one of the most important factors for rethinking work processes and transforming current processes.

It is also important to understand the kinds of individuals who believe in the benefits of a collective information system, such as electronic health records, and those who do not. Managers can benefit from such awareness when developing strategies to introduce upcoming technology and inspire changes in workflow processes. It may be possible for healthcare organisations to take the necessary steps to gain their cooperation by understanding that certain personality types are likely to have a negative attitude towards the introduction of an IS. If advantages are presented to appeal to particular trainees, well-designed and well-structured training could significantly alter the perspectives of such individuals. Using incentive systems to target individuals likely to resist technological advancements is also possible.

It is important to consider all factors contributing to the enhanced and continued use of new technology in organisational change-related management programs. Regarding technological changes within an organisation, it is unlikely that one size fits all. Therefore, when implementing an EHR, it is important to consider each user's personality at every stage.

## 6.2 Limitations and future research

This study only included a population sample from Jordanian public hospitals with fully deployed EHRs; future studies could include private institutions. Nurses were the main subject of study. However, behavioural research by other healthcare professionals regarding the use of CI and EHR for its use would be interesting to study. The survey method often suffers from shortcomings such as the tendency to answer and society's desire to please the researcher, misunderstanding and misinterpretation of the respondents about the questions or wrong answers. The present study used a quantitative approach to investigate nurses' confidence when it comes to using electronic health records, as well as the impact that attitudes and behaviours can have on nurses' Continuance Intention to Use EHRs in this is, indeed, limited. A qualitative technique might elicit deeper insights and provide unconventional support for the outcomes for a greater grasp of the topic. Because the study used a cross-sectional methodology, a longitudinal investigation would be beneficial to corroborate the findings by evaluating factors over time. A cross-analysis of results from different hospital levels is highly recommended for future studies.

The numerous theories in this study consider aspects such as personality critical to management IS research, particularly in health informatics. However, it would also be fascinating to investigate how character affects UTAUT in a larger context. It is fascinating to consider the function of other personality traits in connection to UTAUT. While FFM has stimulated theoretical growth and inspired extensive empirical research in psychology and other domains, research on management IS has generally neglected personality as a crucial aspect, which has been neglected for a long time. The research looked into nurses' Continuance Intention to Use EHR in light of two theoretical frameworks: UTAUT and FFM. It ignores many of the major external factors affecting users after applying the technology, for example, satisfaction, which future studies may investigate. Given the rising interest in the relationship between human-computer interaction and IS research, future assessments should investigate interactions, their consequences and their effects on the continuous usage of IS utilising the framework offered in this study.

## Supporting information

**S1 Data.**
(CSV)

## Author Contributions

**Conceptualization:** Adi Alsyouf, Abdalwali Lutfi, Khalid Anwer Al-Mugheed, Mahmaod Alrawad, Mohammed Amin Almaiah, Disha Sawhney.

**Data curation:** Adi Alsyouf, Nizar Alsubahi.

**Formal analysis:** Adi Alsyouf, Abdalwali Lutfi, Khalid Anwer Al-Mugheed, Mahmaod Alrawad, Mohammed Amin Almaiah.

**Funding acquisition:** Adi Alsyouf, Nizar Alsubahi.

**Investigation:** Adi Alsyouf, Rami J. Anshasi.

**Methodology:** Adi Alsyouf, Nizar Alsubahi, Rami J. Anshasi, Fahad N. Alhazmi.

**Project administration:** Adi Alsyouf, Rami J. Anshasi, Fahad N. Alhazmi.

**Resources:** Adi Alsyouf, Abdalwali Lutfi, Khalid Anwer Al-Mugheed, Mahmaod Alrawad, Mohammed Amin Almaiah.

**Software:** Adi Alsyouf.

**Supervision:** Adi Alsyouf.

**Validation:** Adi Alsyouf, Abdalwali Lutfi, Khalid Anwer Al-Mugheed, Mahmaod Alrawad, Mohammed Amin Almaiah.

**Visualization:** Adi Alsyouf, Haitham Alali.

**Writing – original draft:** Adi Alsyouf, Haitham Alali.

**Writing – review & editing:** Adi Alsyouf, Fahad N. Alhazmi.

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
