## [Decision Letter · Decision Letter 0]

4 Mar 2024

Nurses' Continuance Intention to Use Electronic Health Record Systems The Antecedent Role of Personality and Organisation Support

PONE-D-23-31970

Dear Dr. Alsyouf,

My compliments!

We’re pleased to inform you that your manuscript has been judged scientifically suitable for publication and will be formally accepted for publication once it meets all outstanding technical requirements.

Kind regards,

Sabina De Rosis, PhD

Academic Editor

PLOS ONE

 "Nurses' Continuance Intention to Use Electronic Health Record Systems: The Antecedent Role of Personality and Organization Support

Dear Editor,

PLOS ONE Journal

Subject: Funding Declaration for the Paper Titled "Nurses' Continuance Intention to Use Electronic Health Record Systems: The Antecedent Role of Personality and Organization Support"

I am writing to declare the funding sources that supported our research, which is submitted for your esteemed journal's consideration.

1. I, Adi Alsyouf, have received funding grants from King Abdulaziz University under grant number G: 533-849-1443.

2. My co-author, Nizar Abdulhai Alsubahi, has obtained funding grants in association with Maastricht University.

We believe it is essential to maintain transparency in our research endeavors, and hence, we share this information with PLOS ONE to ensure integrity in the publishing process.

Thank you for your attention to this matter.

Sincerely,

Adi Alsyouf"

Please respond by return e-mail so that we can amend your financial disclosure and competing interests on your behalf.

Additional Editor Comments (optional):

Reviewers' comments:

Reviewer's Responses to Questions

**Comments to the Author**

1. Is the manuscript technically sound, and do the data support the conclusions?

Reviewer #1: Yes

Reviewer #2: Yes

2. Has the statistical analysis been performed appropriately and rigorously? 

Reviewer #1: Yes

Reviewer #2: I Don't Know

3. Have the authors made all data underlying the findings in their manuscript fully available?

Reviewer #1: Yes

Reviewer #2: Yes

4. Is the manuscript presented in an intelligible fashion and written in standard English?

Reviewer #1: Yes

Reviewer #2: Yes

5. Review Comments to the Author

Reviewer #1: I applaude your effort.

Linking UTAUT with nursing is extremely innovative. The importance of this strain of research can prepare new ground for future research in the managerial and organizational field, i.e. nursing management.

Reviewer #2: Thank you for the opportunity to review your manuscript. I found it both interesting and informative. It was interesting to see the overlay and different lens of the personality traits of nurses when either accepting (or not) the continued use of EMR post implementation. Personally, I had never taken into consideration, in such depth, of the personality traits of the multiusers. I did find the introduction and the literature review, while thorough di tend to be a little long. This could be a little more concise. Aims and objectives of the study were readily identified and addressed. Discussion was supported with appropriative literature and reads well. The manuscript was logically sequenced and well written, with minor anomalies, such as commencing a sentence with an acronym, only noted.

6. PLOS authors have the option to publish the peer review history of their article (what does this mean?). If published, this will include your full peer review and any attached files.

Reviewer #1: **Yes: **Chiara Barchielli, Ph.D.

Reviewer #2: No

---

## [Editor Report · Acceptance letter]

20 Sep 2024

PONE-D-23-31970 

PLOS ONE

Dear Dr. Alsyouf, 

I'm pleased to inform you that your manuscript has been deemed suitable for publication in PLOS ONE. Congratulations! Your manuscript is now being handed over to our production team.

Kind regards, 

on behalf of

Dr. Sabina De Rosis 

Academic Editor

PLOS ONE